

# Airborne electromagnetic data leveling based on structured variational method

Qiong Zhang[1,2], Xin Chen[1,2], ZhongHang Ji[1,2], Fei Yan[1,2], ZhengKun Jin[1,2], YunQing Liu[1,2]

[1] School of Electronics and Information Engineering, Changchun University of Science and Technology, Changchun 130022, China

[2] Jilin Provincial Science and Technology Innovation Center of Intelligent Perception and Information Processing, Changchun, Jilin, China

*Correspondence to:* Y. Q. Liu (mzliuyunqing@163.com)

**Abstract.** The leveling errors are defined as the data difference among flight lines in airborne geophysical data. The differences of the signal leveling always show as a striping pattern parallel to the flight lines on the imaged maps. The fixed structured pattern inspires us to structure a guided leveling error model by an anisotropic Gabor filter. Then we embed the leveling error model in total variational framework to flexibly calculate leveling errors. The guided leveling error model constrain the noise term of total variation rather than just blind removal. Moreover, the structured variational method can be extended to remove other type of noises which have general noise priors. We have applied the method to the airborne electromagnetic, magnetic data, and apparent conductivity data collected by Ontario Geological Survey to confirm its validity and robustness by comparing the results with the published data. The structured variational method can better level airborne geophysical data based on the space properties of leveling error.

## 1 Introduction

In airborne geophysical exploration, dynamic measuring mode brings convenience and efficiency, but also constantly changes surrounding environment of the aircraft (Luyendyk, 1997; Gao et al., 2021). The aircraft data are acquired under different flight conditions and have the unequal data levels which defined as leveling errors. Leveling errors showed as the striping pattern along the flight direction because of continuous "S-type" flight mode (Hood, 2007).

Airborne geophysical survey is commonly carried out in a long-term and large-scale measurement. Mathematically, a variety of factors contribute to the leveling errors which are described as distributed parameter model. The uncontrollable external environment is the main source of the leveling errors in airborne geophysical survey. The seasonal and regional climate brings with the temperature fluctuations and natural wind changes. The temperature influences the internal aircraft configuration, and the wind directly changes the external inclination angle of aircraft (Huang and Fraser, 1999; Valleau, 2000; Siemon, 2009). It concludes that the external environment cannot be deemed a lumped parameter model and indirectly affects the data levels of each survey point.

Other factors are related to the intrinsic property of airborne measuring. Airborne survey routinely flies in a continuous "S-type" flight mode under the certain elevation. When the aircraft changes the flying direction, the aircraft sides on the left and right are alternately to face the same surrounding environment. The opposite direction between adjacent lines makes the





minor difference of flight attitude angle and other system configurations (Yin and Fraser, 2004; Huang, 2008). In addition, it is unavoidable to keep constant flying altitude, no matter how advanced systems are and how experienced personnel operate

(Tezkan et al., 2011; Eppelbaum and Mishne, 2011). The minor fluctuation factors are hard to control and measure that contribute to leveling errors.

The sources of the leveling errors are multiplicative and unable to quantitatively describe. It is hard to set up mechanism modelling of leveling errors. Currently, geophysicists base on the definition of leveling error and carry out data processing.

### 1.1 Tie-line leveling method

A traditional but effective method is tie-line leveling. Compared the flight line data with tie line data at the same survey point, the operators correct the crossover point based on the differences of the tie lines and flight lines. The accuracy of tie-line leveling mainly relies on whether the differences can match with the leveling errors. Many geophysicists have proposed algorithms to improve matching precision (Foster et al., 1970; Yarger et al., 1978; Bandy et al., 1990; Mauring et al., 2002; Srimanee et al., 2020). However, the flight line data and the tie line data are flown in different aircraft configuration and

external environment. Moreover, airborne electromagnetic data are relatively sensitive to altitude compared with airborne magnetic data. The leveling error is not the only cause to accumulate the differences of the crossover point. It is hard to separate the leveling errors from differences. Furthermore, virtual tie lines (Huang and Fraser, 1999; Fan et al., 2016; Zhang et al., 2018) are skillfully constructed to level geophysical data instead of tie lines.

### 1.2 Block leveling method

From the definition of leveling error, the inconsistent data level in flight lines is attributed to leveling errors which are not continuous between adjacent flight lines. However, survey area geology changes quite slowly, it is reasonable to assume the nature survey points are correlate in a certain region. Then the leveling errors can be derived line-to-line based on the differences between adjacent flight lines (Green, 2003; Huang, 2008; Zhu et al., 2020). Moreover, geophysicists skilfully constructed one-dimensional (1D) flight line windows and two-dimensional (2D) planar windows, considering the statistical

parameters difference between the flight line data and region data. The leveling errors are calculated point-to-point by matching with the difference between the 1D and 2D window values (Mauring, 2006; Beiki, 2010; Ishihara, 2015). Moreover, the geophysical data can be microleveled using the statistical approach in designed moving window (Davydenko and Grayver, 2014; Groune et al., 2018).

### 1.3 Global leveling method

The line-to-line and point-to-point methods only level small amounts of data in each loop that can be deemed as block processing methods. A common problem is cumulative inaccuracies when the leveled data are used to level in next loop. In contrast, global processing methods operate the entire region data instead of only part data in every iteration. The global processing methods available mainly focus on airborne magnetic data leveling based on the separated long-wavelength





components (Urquhart, 1988; Nelson, 1994; Luo et al., 2012; White et al., 2015; Zhang et al, 2021). The directional filters
are designed and leveled the geophysical data (Minty, 1991; Ferraccioli et al., 1998; Siemon, 2009; Gao et al., 2021).

In summary, the conventional block processing methods would inevitably transfer errors. The global processing methods mainly focus on leveling airborne magnetic data. As the leveling error properties discussed above, the leveling error is an additive drift, presented as the inconsistent data level among the flight lines. The inconsistent is affected by a variety of factors that we are hard to construct the mechanism model of leveling error. But the striping errors would increase the total
variation of the measuring area. Then total variational theory inspires us to leveling data by inducing an energy functional. The proposed method is described as follows.

## 2 Proposed Method

### 2.1 Total Variational Model

The theoretical basis of most leveling techniques is that the geophysical field is continuous. The observed data tend to show
significant correlations with their neighboring points. But the leveling errors are not continuous between adjacent flight lines (Huang, 2008). When the assumption is valid, the geophysical data with leveling errors will have a large variation amplitude, compared with nature geophysical data. Then it is advisable to estimate the leveling error components based on total variation model.

We simply deemed the survey data consists of two parts:

$\qquad \mathbf{S}(i,j) = \mathbf{E}(i,j) + \mathbf{D}(i,j),$ (1)

where $\mathbf{S}(i,j)$ is the $ith$ survey data in the $jth$ flight line, $\mathbf{E}(i,j)$ is the leveling error component of the survey point, $\mathbf{D}(i,j)$ is the leveled data. Here the survey data are considered as a 2D function in entire region $\mathbf{\Omega}$, $(i,j)$ define the $ith$ survey data in the $jth$ flight line.

Rudin, Osher, and Fatem (1992) introduced total variation norm and proposed ROF total variation model which has been
widely used in image-denoising applications. Based on total variational model, we can estimate the leveling error components by constructing an energy functional,

$\qquad F(\mathbf{D}) = \int_{\Omega} \|\mathbf{S} - \mathbf{D}\|^2 + \lambda TV(\mathbf{D}),$ (2)

where $\lambda$ is the regularization coefficient that quantifies the degree of smoothness, $TV(\mathbf{D})$ is the total variation of the estimated solution $\mathbf{D}$ expressed as,

$\qquad TV(\mathbf{D}) = \int_{\Omega} |\nabla \mathbf{D}| = \int_{\Omega} \sqrt{\left(\frac{d\mathbf{D}}{dx}\right)^2 + \left(\frac{d\mathbf{D}}{dy}\right)^2} \, dxdy.$ (3)

In the total variational model, $\int_{\Omega} \|\mathbf{S} - \mathbf{D}\|^2$ is a fidelity term which ensures the similarity between the original data $\mathbf{S}$ and the clear data $\mathbf{D}$. In Eq. (2), $L$-2 norm is selected to build the fidelity term as its excellent edge-preserving performance.



$TV(\mathbf{D})$ serves as the regularization term, aiming at penalizing the undesirable damage in data. The regularization term is the total variation of the estimated solution. It means that gradient-domain sparse constraints are imposing along horizontal and vertical directions. Combining with prior information, the leveling error components can be computed by minimizing total variation model in Eq. (2). Total variational model has applied on the striping noise removal (Zhang and Zhang, 2016; Liu et al., 2019).

## 2.2 Leveling Error Model

Accurate extracting leveling errors requires to combine the total variation model with as much prior information of leveling error as possible. Leveling errors present a significant directional property and showed as the striping pattern along the flight direction. Then we can design an anisotropic Gabor filter with principal axis directed by the leveling error.

In geophysical exploration, the leveling error model should estimate the intensity at each survey point that can be modeled as,

$$\mathbf{E} = \alpha * \mathbf{G}, \tag{4}$$

where $\alpha$ is the weight coefficient which describes the intensities of leveling error, $\mathbf{G}$ is the noise pattern. We model stripes as anisotropic Gaussian function defined by:

$$\mathbf{G}(i,j) = e^{-\frac{x_i^2}{\sigma_i^2}-\frac{y_j^2}{\sigma_j^2}}$$
$$\begin{cases} x_i = i\cos\theta + j\sin\theta \\ y_j = -i\sin\theta + j\cos\theta. \end{cases} \tag{5}$$

In the Eq. (5), $(i,j)$ defines the location of the *ith* survey data in the *jth* flight line. $\theta$ represents the normal's orientation to the Gabor function's parallel stripes, that is, the flight line direction. $\sigma_i$ and $\sigma_j$ are the Gaussian envelope's standard deviation of $x$ direction and flight line direction respectively.

The pattern of leveling error is mainly described by the Gaussian function. We can obtain the parameters in Eq. (5) combined prior shape information. The weight coefficient defines the leveling error intensity that is necessary to solve from the overall view.

## 2.3 Structured Variational Model

When we guide the total variational model leveling by leveling error model, the structured variational model provides an accurate geophysical processing design. We obtain the following objective function:

$$F(\alpha,\lambda) = \int_{\Omega}\|\alpha * \mathbf{G}\|^2 + \lambda TV(\mathbf{S} - \alpha * \mathbf{G}). \tag{6}$$

Equation (6) contains two coefficients $\alpha$ and $\lambda$ to balance the fidelity term and regularization term. It is permitted to reasonably merge the two coefficients and express Eq. (6) as,



$$F(\alpha) = \int_{\Omega} \|\alpha * \mathbf{G}\|^2 + TV(\mathbf{S} - \alpha * \mathbf{G}). \tag{7}$$

Then we use alternating direction method of multipliers (ADMM) to solve nonconvex optimization problems. ADMM converts the original problem into subproblems with closed-form solutions. It is an effective approach in a sequence of

iterative sub-optimizations (Bertsekas, 1982).

While the leveling error intensity for each survey point is solved, we complete the data leveling using Eq. (1) and Eq. (4) under the structured variational model.

In exploration field, airborne geophysical measurement data contains a large amount of noise due to atmospheric flow, lightning, aircraft vibration, and unstable speed factors (Yin C. C. et al, 2015). In addition to leveling errors, different

kinds of noises damage the measurement data simultaneously. Here, we simply assume the measurement data contains leveling errors and Gaussian white noise. In that case, the proposed leveling method has an obvious advantage compared with other leveling methods. The proposed method constructs an energy functional as Eq. (2). For other noise, we can consider the denoising problem under the framework similarly. The noise model in Eq. (4) describes the noise distribution and geometrical structure. When we try to remove several kinds of several kinds of disturbances, Eq.

(4) is extended as,

$$\mathbf{E} = \sum_{i=1}^{n} \alpha_i * \mathbf{G}_i, \tag{8}$$

where *n* is the number of noise type.

For Gaussian white noise, it can be obtained by convolving a Dirac function with a sample of white Gaussian noise. The proposed method simultaneously removes the leveling errors and Gaussian white noise in one step processing

which helps to improve electromagnetic exploration accuracy.

Thus, there are three evident advantages in proposed leveling method:

(1) Total variation model designs the total energy as a constraint condition and obtains the constrained gradient minimization by the regularization coefficient. When we use total variational model to deal with survey area data, it can reasonably remove the leveling errors that increase the gradient of survey area data.

(2) Due to the complexity of airborne geophysical field measurement, there are multiple components in airborne geophysical data. To focus on leveling error extracting, we construct a rough leveling error model based on the striping pattern. Then the leveling error model is embedding into the gradient minimization functional and clearly solved in the structured variational model.

(3) The structured variational model can be carried over into other noise. If it is accessed the noise characteristic and

established the noise model, we can speculate that the structured variational model can remove other noises. The framework may take effect based on precise noise model.

We have verified the advantages through experiments.



## 3. Results

### 3.1 Airborne magnetic data leveling

**3.1.1 Real dataset example**

The leveling method has been tested on magnetic field data obtained by Geotech Limited. Figure 1 shows the magnetic data before and after leveling. The survey area data include 117 flight lines with a line spacing of 200 m and contain striped leveling errors along flight-line direction. In the example, we only focus on leveling errors in Fig. 1(a). The noise pattern in Eq. (5) is set based on the prior information about leveling errors. For example, we set the normal's orientation $\theta$ as 90°

because the flight line direction is vertical in the general coordinate system. The Gaussian envelope's standard deviation $\sigma_i$ and $\sigma_j$ decide the number of stripes. And the ratio of $\sigma_i$ and $\sigma_j$ represents the spatial shape of the Gabor function. When we use the Gabor function to describe leveling errors, $\sigma_i/\sigma_j$ should be much less than 1. Figure. 1(b) shows the processed data by the proposed leveling method. And Fig. 1(c) presents the leveled data by the classic Tie-line leveling method.

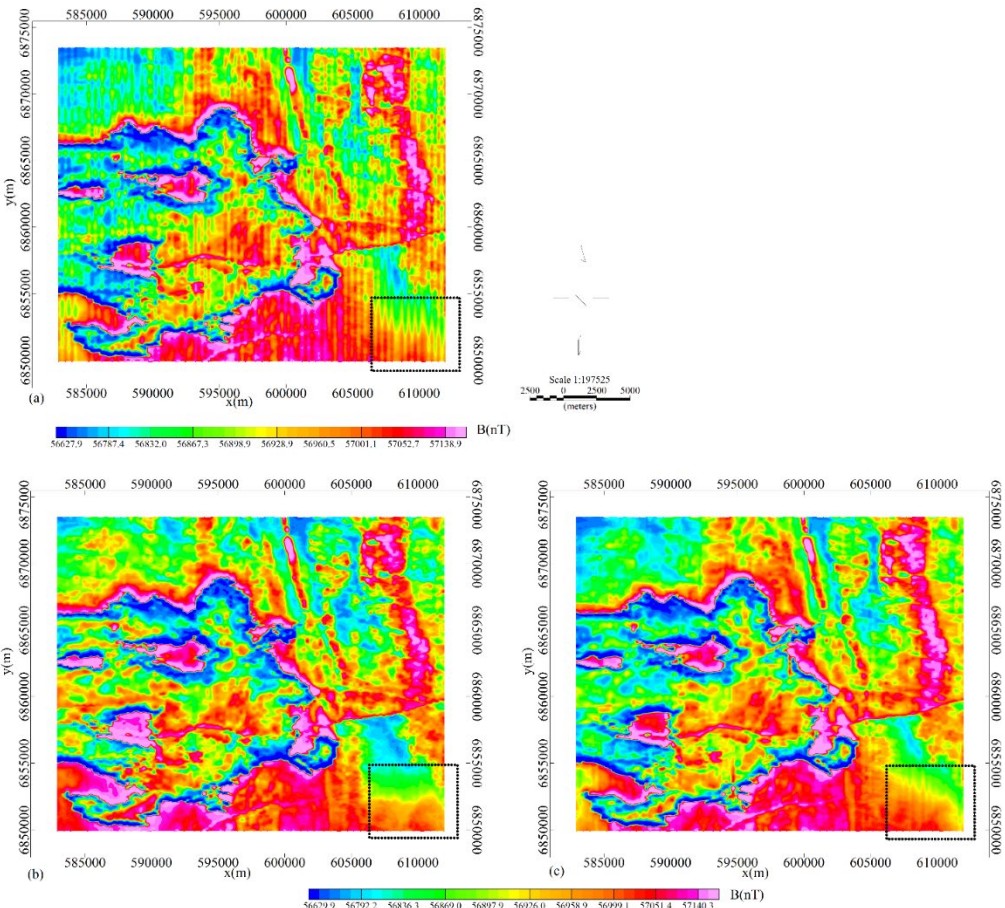

**Figure 1: Airborne magnetic data leveling. (a) Raw magnetic data. (b) Leveling results by the proposed leveling method. (c) Leveling results by Tie-line leveling method.**





### 3.1.2 Synthetic dataset example

The example is from a synthetic magnetic dataset with additional Gaussian white noise and leveling errors. We selected the leveling results by Tie-line leveling method as the clean data. The data have been explained in Real dataset example and

presented in Fig. 1(c). Then we tested our algorithm on the noisy magnetic data as Fig. 2 and Fig.3 shown. There are three experiments, including specific clean data with Gaussian white noises, clean data with leveling errors, and clean data with Gaussian white noises and leveling errors.

The first synthetic dataset focuses on removing Gaussian white noises. We required to estimate and obtain the noise model by white noise estimation method. Then the structured variational model will be guided to remove the corresponding noise

type. Figures 2(a) and 2(d) show the data before and after processing. The second synthetic dataset focuses on removing leveling errors. Figure 2(b) is the clean data with leveling errors. We use a Gabor filter to simulate the leveling error model. And Fig. 2(e) show the data after processing. The third synthetic dataset is designed with two noise components as Fig. 2(c) shown. While the proposed method intends to remove the noises simultaneously, the noise model in Eq. (8) must include Gaussian white noise model and leveling error model. Then the proposed method removes the two noises in an objective

function. Figures 2(c) and 2(f) show the noisy magnetic data before and after processing.









**Figure 2: Synthetic airborne magnetic data processing. (a) Magnetic data with Gaussian white noises. (b) Magnetic data with leveling errors. (c) Magnetic data with Gaussian white noises and leveling errors. (d) Denoised results of Fig.(a) data. (e) Denoised results of Fig.(b) data. (f) Denoised results of Fig.(c) data. (a), (b), and (c) have been adjusted to the same colorbar. (d), (e), and (f) have been adjusted to the same colorbar.**

Furthermore, we calculated the signal to noise ratio (SNR) for the three experiments. The quantitative comparison is shown in Tab. 1. And Fig. 3 illustrates the transient data to compare the results in greater detail. There are four flight lines locally enlarged, corresponding to the black dotted rectangle in Fig. 2. The three subgraphs analyzed separately the three experiments above. In every subgraph, the blue curve represents the clean magnetic data, the red curve represents the noisy magnetic data, and green curve represents the denoised magnetic data.

**Table 1: SNR of synthetic airborne magnetic data processing**

|  | Gaussian white noises | Leveling errors | Gaussian white noises and leveling errors |
|---|---|---|---|
| Before data processing | 65.62 dB | 51.44 dB | 51.40 dB |
| After data processing | 72.34 dB | 75.94 dB | 65.30 dB |

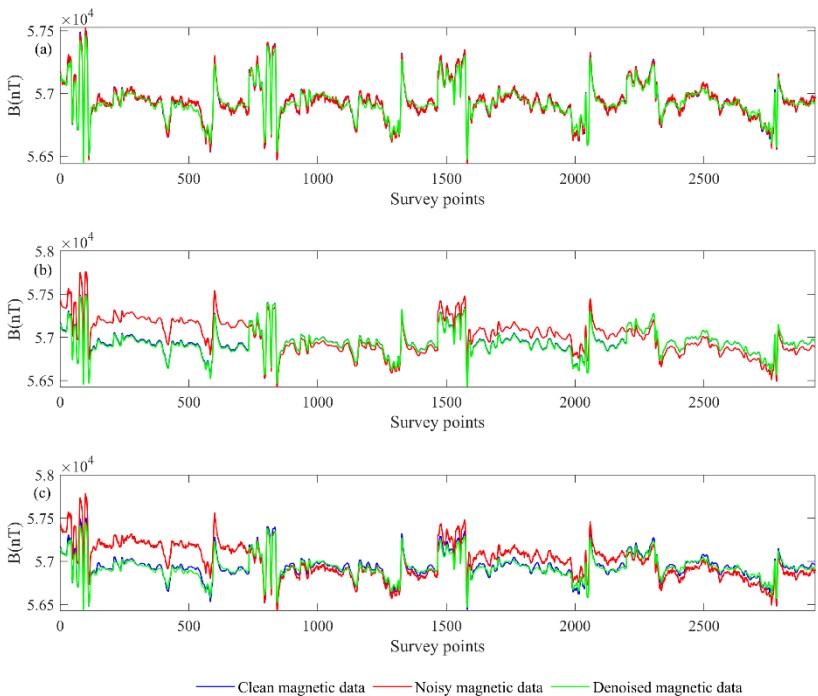

**Figure 3: Leveling result analysis of synthetic airborne magnetic data. (a) Magnetic data with Gaussian white noises. (b) Magnetic data with leveling errors. (c) Magnetic data with Gaussian white noises and leveling errors.**

## 3.2 Apparent conductivity data leveling

We also tested the leveling method on the apparent conductivity data provided by Ontario Airborne Geophysical



Surveys. The dataset used in the paper is formed by 70 flight lines named L310-L1000 as a part of Geophysical Data Set 1076 measured in the surveys (Ontario Geological Survey, 2014). The apparent conductivity data are calculated

from dBz/dt response at 97 m average depth from the surface. The key transformation algorithm is based on scheme of the apparent resistivity transform of Maxwell A. Meju (1998).

Figure 4 presents the apparent conductivity data before and after leveling processing. As Fig. 4(a) presented, there are only slight striped errors along the flight line direction in the apparent conductivity data. While the electromagnetic data are transformed into conductivity parameters, the altitude sensitivity is weakened strongly (Fraser, 1972; Huang

and Fraser, 1999).

Then we applied the structured variational model to the apparent conductivity data and got the leveling results as Fig. 4(b) shown. In the analysis of the data, it is assumed that the leveling error is only noise source. Figure 5 illustrates the transient data to compare the results in greater detail. Two part data are plotted corresponding to the black rectangle in Fig. 4. The first part including 10 flight line data as shown in Fig.5 (a). Then we selected a small data scope (0-2.4*10$^{-3}$

S/m) to locally enlarge and drawn the data in Fig.5 (b). Figures 5(c) and 5(d) are drawn by the 5 flight line data which are corresponding to the black dash-dot rectangle in Fig.4.

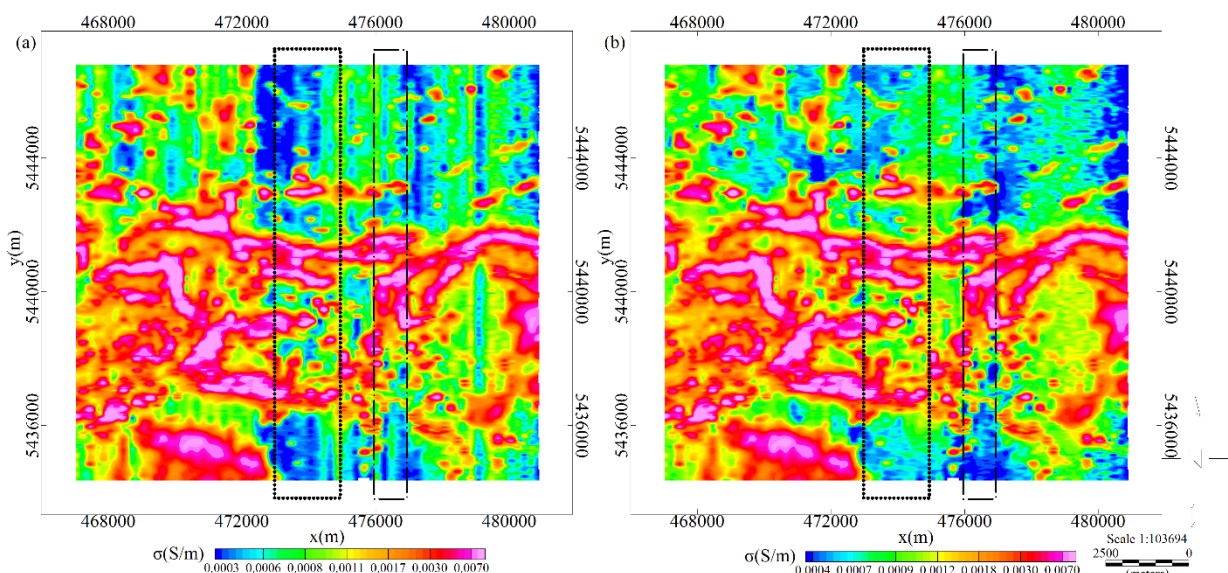

**Figure 4: The leveling of the apparent conductivity data. (a) The raw data. (b) Leveling results by the proposed leveling method.**





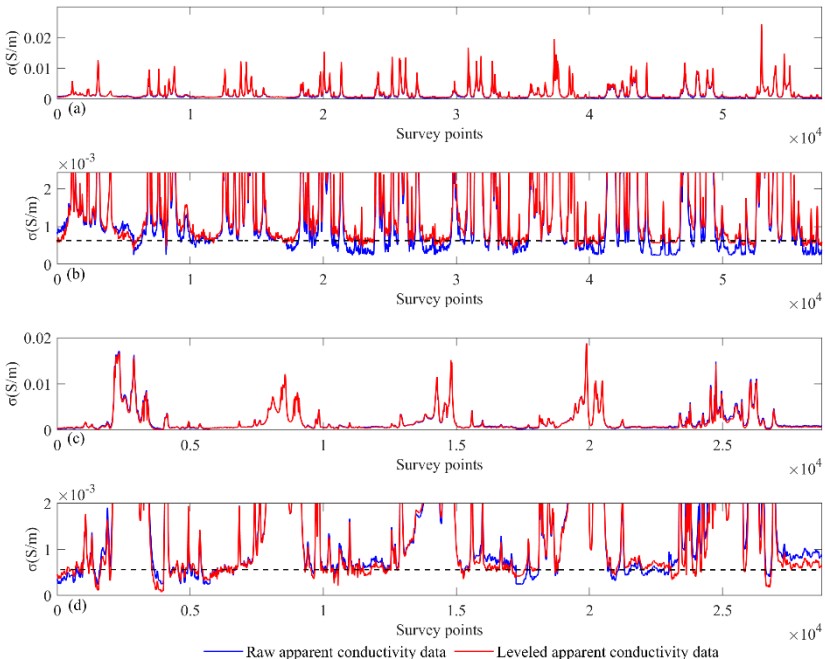

**Figure 5: Leveled apparent conductivity data. (a) 10 flight line data, corresponding to the black dotted rectangle in Fig.4. (b) Local enlarged curves of Fig.(a) data. (c) 5 flight line data, corresponding to the black dash-dot rectangle in Fig.4. (d) Local enlarged curves of Fig.(c) data.**

## 4 Discussions

Firstly, we analyzed and discussed the leveling results in airborne magnetic data example in Fig. 1. As seen in Fig. 1(b) and (c), most of the striped leveling errors have been removed by the proposed leveling method and Tie-line leveling method. Careful contrast of the two results shows Tie-line leveling method remains some weak leveling errors which is clear in the black dotted bordered rectangle in Fig. 1.

The residues in Tie-line leveling method may be caused by the incompatible data alignment. Although the leveling errors show striped pattern in survey area map, they are slowly changing from point to point in certain flight line. Tie-line leveling method adjusts the flight line data to match tie line data. Because tie line number is much less than point survey number, it needs to build a model by the crossover point differences of the tie lines and flight lines. When a few tie line data are used to calculate the leveling error of every point, it is hard to balance every point by an exact model.

In the paper we proposed a new technology based on the ROF total variation model which focuses on the gradient change of measured data. As the basic principle of data leveling, theoretical geophysical data have continuous change regularities. And leveling errors break the continuity and increase the total variation of survey area data (Zhang et al, 2022). In the proposed leveling method, the structured variational model aims at minimizing the energy functional that can better explore the leveling errors in the data.





Then we evaluated a synthetic magnetic example to further analyze the results. There are three experiments with different noises: (1) Gaussian white noises; (2) leveling errors; (3) mixed noises with Gaussian white noises and leveling errors. As shown in Fig. 2, the structured variational model can visibly remove the noises. In theory, noises increase the gradient amplitude. The proposed model can be robust to smooth the gradient of survey area data in an energy functional.

There is a transient data comparison in Fig. 3. In the three experiments, the results of (green lines) are highly similar to clean data (blue lines). And the data processing is without the localized anomalies being trimmed. Three groups of SNR are calculated in Tab.1. The robustness of proposed model means it can deal with different noise type. Suitable noise model still needs to be set, otherwise, it may lead to over-smoothing effect.

Finally, we test the leveling method on apparent conductivity data. Compared with Figs. 1(a) and 4(a), apparent conductivity as response domain are slightly effected by leveling errors. However, the differences in data levels still interfere data interpretation. Figure 5 can better evaluate how well the method is working. We deem the bottom of data curve represents the data level and enlarge the small data scope as Figs. 5(b) and 5(d) shown. And a black dashed line is added as a measured rule. The blue lines in Fig. 5 are the apparent conductivity data without leveling. It is obvious that the bottom of blue lines hover around the black dashed lines in Figs. 5(b) and 5(d). When we adjusted the data, the data levels are united as the red lines shown in Fig. 5. The slight leveling errors are tested and removed by the proposed leveling method. The method is effective to time-domain airborne electromagnetic data and response-domain airborne electromagnetic data.

## 5 Conclusions

In this paper, we proposed a leveling method based on a structured variational method. The basis is that leveling errors increase the gradient of survey area data. The ROF total variation model is proposed by Rudin, Osher, and Fatem (1992) and designed with the total energy as a constraint condition. Moreover, it has a potential performance in smoothing the total gradient by minimizing the constrained gradient. The regularization coefficient plays a role in controlling the smoothness. The ROF total variation model can adjust the airborne electromagnetic data by smoothing the total gradient.

A rough leveling error model is constructed to focus on leveling error accurately. Based on the leveling error characteristic, we introduced the Gabor filter to match the leveling error with the striping pattern. Furthermore, the rough leveling error model is embedded into the ROF total variation model to construct a structured variational model. The proposed model is guided to deal with the additional gradient caused by leveling errors. We have confirmed the method's reliability by applying it to the magnetic, synthetic magnetic, and apparent conductivity data.

Besides, the synthetic magnetic example has tested the structured variational model, which can also handle other noise. A suitable noise model still needs to be embedded into the ROF total variation model. Otherwise, it may lead to an over-smoothing effect and loss of accuracy.



**Code/Data availability**

The data used in the paper have been opened by Ontario Geological Survey. The more information can be found in the official website (https://www.mndm.gov.on.ca/en).

**Author contribution**

The manuscript is approved by all authors for publication. Zhang and Chen developed the algorithm model and performed the simulations. Ji designed the experiments. Jin and Yan carried them out. Liu prepared the manuscript with contributions from all co-authors.

**Competing interests**

The authors declare that they have no conflict of interest. I declare on behalf of my co-authors that the work described was original research that has not been published previously and is not under consideration for publication elsewhere, in whole or in part.

**Acknowledgements**

This study was supported by the National Natural Science Foundation of China "Research on Key Technologies of Airborne Electromagnetic Data Leveling Method (42204144)".

We all thank the Ontario Geological Survey for permission to use the geophysical data in this study. Moreover, the authors are appreciative of Jean M. Legault, Chief Geophysicist of Geotech Ltd, for providing us with access to Geophysical Data Set 1076. We also appreciate editors and reviewers very much for their positive and constructive comments and suggestions on our manuscript.

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
