# Peer review of "Airborne electromagnetic data leveling based on structured variational method"

_EGUsphere, 2024_

## Author Response (AR1)

**Response to Reviewer:**

Many thanks for your significant comments on our paper. We have revised the manuscript according to your comments. The response to each revision is listed as following:

**1. Comment 1:**

The background theories should be provided in details. They are insufficient to describe the research.

**Response:** As the reviewer suggested, we have modified the manuscript. The supplemented introductions are given as below shown.

**Original version:** In airborne geophysical exploration, dynamic measuring mode brings convenience and efficiency, but also constantly changes surrounding environment of the aircraft (Luyendyk, 1997; Gao et al., 2021). The aircraft data are acquired under different flight conditions and have the unequal data levels which defined as leveling errors. Leveling errors showed as the striping pattern along the flight direction because of continuous "S-type" flight mode (Hood, 2007).

**Modified version:** Airborne geophysical exploration is loaded on an aircraft which moves at a high speed and at a certain elevation. The dynamic measuring mode brings convenience and efficiency, but also constantly changes surrounding environment of the aircraft (Luyendyk, 1997; Gao et al., 2021). The aircraft data are acquired under different flight conditions and have the unequal data levels which defined as leveling errors. Leveling errors showed as the striping pattern along the flight direction because of continuous "S-type" flight mode (Hood, 2007).

A revised manuscript with the marked correction was attached as the supplemental material entitled by "track-changes-file".

**2. Comment 2:**

Please enhance the overall presentation of the research. Figures are not informative enough and the equations are not formed scientifically. Data levelling should be explained formed properly.

**Response:** As the reviewer suggested, we have modified the manuscript.

We have added short introductions at each section for readability, especially in the part of "Proposed Method". The additional introduction is shown as below.

As the survey area space analysis, leveling errors are formed along with the flight lines and have definitive directional distribution property (Zhang, 2022). The directional stripes would further cause the discontinuity from the vertical direction and increase the horizontal gradient amplitude. Total variational model can detect and remove all the components which impair the total smoothness. While we specifically focus on leveling errors, a detailed constraint is helpful. So, we build a leveling error model based on its prior information and properly embed the model in the total variational model. In the proposed method, only the leveling errors are extracted and removed through solving the constrained and structured variational model.

A revised manuscript with the marked correction was attached as the supplemental material entitled by "track-changes-file".

**3. Comment 3:**

Please check the statement "structured variation method can be extended to remove other type of noises".

**Response:** We have modified the manuscript as the reviewer suggested. The detailed change is listed as following.

**Original version:** Moreover, the structured variational method can be extended to remove other type of noises which have general noise priors.

**Modified version:** Moreover, we can also apply the structured variational method to remove other noises in airborne geophysical data. It would just require replacing the noise prior models in the proposed method.

**4. Comment 4:**

Please discuss the data collection, processing and analyses processes.

**Response:** As the reviewer suggested, we have modified the manuscript. In apparent conductivity data leveling example, we have added the introduction of the data in the latest manuscript.

**Original version:** We also tested the leveling method on the apparent conductivity data provided by Ontario Airborne Geophysical Surveys. The dataset used in the paper is formed by 70 flight lines named L310-L1000 as a part of Geophysical Data Set 1076 measured in the surveys (Ontario Geological Survey, 2014). The apparent conductivity data are calculated from dBz/dt response at 97 m average depth from the surface. The key transformation algorithm is based on scheme of the apparent resistivity transform of Maxwell A. Meju (1998).

**Modified version:** We also tested the leveling method on the apparent conductivity data provided by Ontario Airborne Geophysical Surveys. The dataset used in the paper is formed by 70 flight lines named L310-L1000 as a part of Geophysical Data Set 1076 measured in the surveys (Ontario Geological Survey, 2014). Geotech Limited carried out a helicopter-borne combined aeromagnetic and electromagnetic survey for the Ministry of Northern Development and Mines in 2014 in the Nestor Falls area in north-western Ontario. Based on Resistivity depth imaging (RDI) technique (Meju, 1998), Geotech Limited converted the EM profile decay data into an equivalent resistivity versus depth cross-section, by deconvolution of the measured TEM data. Data compilation and processing were carried out using Geosoft® OASIS montaj™ and programs proprietary to Geotech Ltd (Ontario Geological Survey 2014).

**5. Comment 5:**

Abstract of the paper should state the importance, gaps, method and the key results of the paper. Introduction is long and raw. It should analyse the recent and related works with their advantages and disadvantages. The gaps should be clear and the claimed contributions should be well justified. Paragraphs are too short, not connected properly and sections are too large and therefore the paper is too long for an academic paper.

**Response:** As the reviewer suggested, we have modified the manuscript. A revised

manuscript with the red marked correction was attached as the supplemental material entitled by "track-changes-file".

**6. Comment 6:**

Please note that recently advanced model-free, uncertain artificial intelligence-based optimal planning approaches for the uncertain air systems are developed which should be addressed. One can see this recent and related one: Minimum distance and minimum time optimal path planning with bioinspired machine learning algorithms for faulty unmanned air vehicles.

**Response:** As the reviewer suggested, we have studied the optimal planning approaches. However, we are currently unable to combine the advanced method with data leveling. Now we also research physics driven and data-driven deep learning methods for scientific computing problems. In our next plans, we would try to introduce physics-informed machine learning to process airborne geophysical data.

**7. Comment 7:**

Please enhance the equations, they are too large and their notations are complex. Incorporating brief insights about them can help their understanding. Please also provide brief results of the research.

**Response:** In the manuscript, we model the leveling errors as stripes along the flight line direction. Equation (5) defined the leveling error stripes as anisotropic Gaussian functions. Then we combine the total variational model in Eq. (2) with the leveling error model in Eq. (4) and Eq. (5). The designed model is presented in Eq. (6) used to calculate the leveling errors.

**8. Comment 8:**

Please state how the regularization parameter lambda is determined.

**Response:** In total variational method, the regularization coefficient $\lambda$ decides the effect on the smoothness of the leveling results. We introduced the spatially adaptive multi-scale model to iteratively decompose the leveling errors which effectively avoid the difficulty on the parameter selection. The method has been published in our other paper as following (Zhang, 2022).

Based on the multiscale hierarchical decomposition theory (Tadmor, 2003), we add the spatially adaptive multi-scale model into the energy functional to avoid the difficulty on the selection of regularization coefficient.

But this manuscript really doesn't cover how to determine an appropriate regularization coefficient. We have added the relevant content in the latest manuscript.

References:
[1]. Luyendyk, A.P.J.: Processing of airborne magnetic data, AGSO Journal of Australian Geology & Geophysics, 17, 31-38, 1997.
[2]. Gao, L. Q., Yin, C. C., Wang, N., Liu, Y. H., Su, Y., and Xiong, B.: Leveling of airborne electromagnetic data based on curvelet transform, Chinese Journal of Geophysics, 5, 1785-1796, doi:10.6038/cjg2021O0089, 2021.

[3]. Hood, P.: History of Aeromagnetic Survey in Canada, The Leading Edge, 26, 1384-1392, doi:10.1190/1.2805759, 2007.

[4]. Zhang, Q., Sun C. C., Yan F., Lv C., Yunqing Liu Y. Q.: Leveling airborne geophysical data using a unidirectional variational model [J]. Geoscientific Instrumentation, Methods and Data Systems, 2022. DOI: 10.5194/gi-2021-33.

[5]. Ontario Geological Survey 2014. Ontario airborne geophysical surveys, magnetic and electromagnetic data, grid and profile data (ASCII and Geosoft® formats) and vector data, Nestor Falls area; Ontario Geological Survey, Geophysical Data Set 1076.

[6]. Meju, M. A.: Short Note: A simple method of transient electromagnetic data analysis, Geophysics, 63, 405-410, doi:10.1190/1.1444340, 1998.

[7]. Tadmor, E., Nezzar, S., and Vese, L.: A multiscale image representation using hierarchical, Multiscale Model Simul, 2, 554-579, doi:10.1137/030600448, 2003.

We tried our best to improve the manuscript and made some revisions in the manuscript. These revisions will not influence the content and framework of the paper. We appreciate for Editors and Reviewers' warm work earnestly, and hope that the correction will meet with approval. Once again, thank you very much for your comments and suggestions.

---

## Author Response (AR2)

**Response to Associate editor:**

There is an inappropriate colour scheme when we check our figures using the Coblis. So we have revised colour scheme in Fig.3 as following:

**Original version:**

[Figure]

**Modified version:**

[Figure]

The revision will not influence the content and framework of the paper.

We appreciate for Editors and Reviewers' warm work earnestly, and hope that the correction will meet with approval. Once again, thank you very much for your comments and suggestions.